# Accurate Thermodynamic Properties of Ideal Bosons in a Highly Anisotropic 2D Harmonic Potential

**DOI:** 10.3390/e25111513

**Published:** 2023-11-03

**Authors:** Ze Cheng

**Affiliations:** School of Physics, Huazhong University of Science and Technology, Wuhan 430074, China; zcheng@mail.hust.edu.cn

**Keywords:** rigorous results in statistical mechanics, Bose–Einstein condensation, cold atoms, 05.30.Jp, 03.75.Hh, 64.60.-i, 67.85.-d

## Abstract

One can derive an analytic result for the issue of Bose–Einstein condensation (BEC) in anisotropic 2D harmonic traps. We find that the number of uncondensed bosons is represented by an analytic function, which includes a series expansion of *q*-digamma functions in mathematics. One can utilize this analytic result to evaluate various thermodynamic functions of ideal bosons in 2D anisotropic harmonic traps. The first major discovery is that the internal energy of a finite number of ideal bosons is a monotonically increasing function of anisotropy parameter *p*. The second major discovery is that, when p≥0.5, the changing with temperature of the heat capacity of a finite number of ideal bosons possesses the maximum value, which happens at critical temperature $T_{c}$. The third major discovery is that, when 0.1≤p<0.5, the changing with temperature of the heat capacity of a finite number of ideal bosons possesses an inflection point, but when p<0.1, the inflection point disappears. The fourth major discovery is that, in the thermodynamic limit, at $T_{c}$ and when p≥0.5, the heat capacity at constant number reveals a cusp singularity, which resembles the λ-transition of liquid helium-4. The fifth major discovery is that, in comparison to 2D isotropic harmonic traps (p=1), the singular peak of the specific heat becomes very gentle when *p* is lowered.

## 1. Introduction

The Bose–Einstein condensation (BEC) of boson systems is always a hot-spot of research in physics. Three independent teams realized the BEC of trapped ultra-cold alkali atoms in 1995 [1,2,3]. Quasi-particles of the Bose type in solids can also be in the BEC state. Recently, scientists have also observed the BEC of excitons [4,5], exciton–polaritons [6,7], and magnons [8,9] in several solid-state systems. Furthermore, the BEC of photons in low-dimensional optical microcavities has become a hot-spot of research in optics [10]. Thermalization of a 2D photon gas has been realized in a dye-filled curved-mirror microcavity [11]. The BEC of 2D photons in a dye-filled curved-mirror microcavity has been observed by Weitz and colleagues [12,13]. In a previous work [14], we have investigated the accurate thermodynamic properties of ideal bosons in a 2D isotropic harmonic potential. In another previous work [15], we have proposed an exact analytic result for ideal boson gases in a highly anisotropic 2D harmonic potential. In this paper, we shall study the thermodynamic properties of ideal Bose atoms in a 2D anisotropic harmonic potential. The exact thermodynamic theory of the BEC state of such ideal Bose atoms will be expounded.

In 1995, the BEC of ultra-cold alkali atoms was realized in 3D harmonic potentials. This achievement kindled great interest in the BEC of ultra-cold bosons in 2D harmonic potentials. The 2D quantum system has been an attractive object since the creation of quantum theory [16]. The 2D characters bring about unbelievably rich phenomena. Some accurate analytic results in 2D systems can be acquired using special methods. First, Salasnich analyzed the thermodynamics of ideal gases in a generic power-law potential [17]. Recently, Weitz and colleagues investigated both experimentally and theoretically the statistical mechanics of a gas of massive photons in a 2D harmonic potential [12,18]. Further, Stein and Pelster investigated the thermodynamic properties of trapped ideal photons in dimensional crossover from 2D to 1D [19]. However, these authors did not compute the internal energy, the entropy, the Helmholtz free energy, and the heat capacity of ideal photons in 2D anisotropic harmonic traps. The Berezinskii–Kosterlitz–Thouless (BKT) transition was realized in ultra-cold Bose atoms in a homogeneous quasi-2D optical trap [20]. For a homogeneous 2D boson system with interactions, either a BEC or a BKT transition occurs depending upon whether the interaction strength is small or large, respectively. In fact, a crossover between BEC and BKT occurs for increasing interaction strength [21].

The object under study is an ideal boson gas in a 2D anisotropic harmonic potential. In reality, a non-interacting Bose–Einstein condensate can be produced by means of Feshbach resonance [22]. In order to deal with an ideal boson gas in a 2D anisotropic harmonic potential, in all the references [23], researchers have utilized various approximate methods. In order to keep off these approximate methods, we propose an exact analytic result in an ideal boson gas in a 2D anisotropic harmonic potential. Using this accurate analytic solution, we elaborate upon the accurate thermodynamic behavior of an ideal boson gas in a 2D anisotropic harmonic potential. This thermodynamic theory is effective for arbitrary temperature, boson number, and anisotropy parameter. The first major discovery is that the internal energy of a finite number of ideal bosons is a monotonically increasing function of the anisotropy parameter *p*. The second major discovery is that, when p≥0.5, the changing with temperature of the heat capacity of a finite number of ideal bosons possesses the maximum value, which happens at critical temperature $T_{c}$. The third major discovery is that, when 0.1≤p<0.5, the changing with temperature of the heat capacity of a finite number of ideal bosons possesses an inflection point, but when p<0.1, the inflection point disappears. The fourth major discovery is that, in the thermodynamic limit, at $T_{c}$ and when p≥0.5, the heat capacity at constant number reveals a cusp singularity, which resembles the λ-transition of liquid helium-4. The fifth major discovery is that, in comparison to 2D isotropic harmonic traps (p=1), the singular peak of the specific heat becomes very gentle when *p* is lowered. These accurate thermodynamic properties of ideal boson gases in a 2D anisotropic harmonic potential can be confirmed in current physics laboratories.

The remainder of this paper is arranged as follows. In Section 2, one derives the thermodynamic potential of ideal boson gases in a 2D anisotropic harmonic potential. In Section 3, we describe the exact thermodynamic behavior of ideal boson gases in a 2D anisotropic harmonic trap. Section 4 depicts the phase transitions of ideal boson gases in the thermodynamic limit. The general conclusions are given in Section 5.

## 2. Thermodynamic Potential of the Ideal Boson System

### 2.1. Many-Particle State in the Particle-Number Representation

The present paper deals with an ideal boson gas with zero spin. The model system is comprised of *N* non-interacting bosons moving in a 2D anisotropic harmonic trap [24]. In the beginning, one must bring out the two operators ${\hat{b}}_{n_{x}n_{y}}$ and ${\hat{b}}_{n_{x}n_{y}}^{†}$. ${\hat{b}}_{n_{x}n_{y}}$ and ${\hat{b}}_{n_{x}n_{y}}^{†}$ denote the destruction and production operators of an atom occupying the oscillator level with $n_{x}$ and $n_{y}$, respectively. ${\hat{b}}_{n_{x}n_{y}}$ and ${\hat{b}}_{n_{x}n_{y}}^{†}$ obey the Bose commutation relations:(1)$$\left[{\hat{b}}_{n_{x}n_{y}},{\hat{b}}_{n_{x}^{\prime }n_{y}^{\prime }}^{†}\right]=\delta_{n_{x}n_{x}^{\prime }}\delta_{n_{y}n_{y}^{\prime }}\;,\;\left[{\hat{b}}_{n_{x}n_{y}},{\hat{b}}_{n_{x}^{\prime }n_{y}^{\prime }}\right]=0\;.$$

Here, we can bring out the number operator ${\hat{N}}_{n_{x}n_{y}}$ of atoms occupying the oscillator level with $n_{x}$ and $n_{y}$ by
(2)$${\hat{N}}_{n_{x}n_{y}}={\hat{b}}_{n_{x}n_{y}}^{†}{\hat{b}}_{n_{x}n_{y}}\;.$$

The eigenvectors of ${\hat{N}}_{n_{x}n_{y}}$ are $|N_{n_{x}n_{y}}\rangle$ and ${\hat{N}}_{n_{x}n_{y}}$ obeys the eigenvalue equation
(3)$${\hat{N}}_{n_{x}n_{y}}|N_{n_{x}n_{y}}\rangle =N_{n_{x}n_{y}}|N_{n_{x}n_{y}}\rangle \;,\;N_{n_{x}n_{y}}=0,1,2,\cdots \;,$$
where $N_{n_{x}n_{y}}$ denotes an eigenvalue of ${\hat{N}}_{n_{x}n_{y}}$. The eigenvector $|N_{n_{x}n_{y}}\rangle$ can be represented as
(4)$$|N_{n_{x}n_{y}}\rangle =\frac{1}{\sqrt{N_{n_{x}n_{y}}!}}{\left({\hat{b}}_{n_{x}n_{y}}^{†}\right)}^{N_{n_{x}n_{y}}}|0\rangle \;.$$

Hence, the total number operator $\hat{N}$ of bosons is presented as
(5)$$\hat{N}=\sum_{n_{x}=0}^{\infty }\sum_{n_{y}=0}^{\infty }{\hat{b}}_{n_{x}n_{y}}^{†}{\hat{b}}_{n_{x}n_{y}}\;.$$

According to the program of second quantization, we can immediately obtain the Hamiltonian of the boson system as
(6)$$\hat{H}=\sum_{n_{x}=0}^{\infty }\sum_{n_{y}=0}^{\infty }E_{n_{x}n_{y}}{\hat{b}}_{n_{x}n_{y}}^{†}{\hat{b}}_{n_{x}n_{y}}\;,$$
where $E_{n_{x}n_{y}}$ is the level of a 2D anisotropic harmonic oscillator given by
(7)$$E_{n_{x}n_{y}}=\left(n_{x}+\frac{1}{2}\right)\hbar \omega_{x}+\left(n_{x}+\frac{1}{2}\right)\hbar \omega_{x}\;,$$
where $\omega_{x}$ and $\omega_{y}$ signify the angular frequencies of the trap along the *x* and *y* axes, respectively. Because the number operator $\hat{N}$ commutes with the Hamiltonian $\hat{H}$, the common eigenstates of $\hat{N}$ and $\hat{H}$ are presented as
(8)$$|\left\{N_{n_{x}n_{y}}\right\}\rangle =\prod_{n_{x}=0}^{\infty }\prod_{n_{y}=0}^{\infty }\frac{1}{\sqrt{N_{n_{x}n_{y}}!}}{\left({\hat{b}}_{n_{x}n_{y}}^{†}\right)}^{N_{n_{x}n_{y}}}|0\rangle \;.$$

The eigenstate vector in Equation (Equation 8) is symmetric under the interchange of any two production operators, consistently with the Bose–Einstein statistics.

### 2.2. Thermodynamic Potential

The eigenstate vector in Equation (Equation 8) represents a multimode number state of bosons, which is a pure state and hence far from thermal equilibrium. Nevertheless, the boson system in a 2D anisotropic harmonic potential goes into thermal equilibrium. This equilibrium is constructed by way of the successive collisions between bosons. The boson system in thermal equilibrium is represented by a definite temperature *T*. Since the number of bosons in the potential is conserved, the 2D boson system has a non-vanishing chemical potential μ. In order to represent the thermal equilibrium state of the boson system, one must devise a grand canonical ensemble of bosons.

In the beginning, one needs to derive the thermodynamic potential Ω. Ω is only a function of temperature and chemical potential. In the grand canonical ensemble at temperature *T*, the grand partition function *Z* is given by
(9)$$Z=Tr\;\exp[-\left(\hat{H}-\mu \hat{N}\right)/k_{B}T]\;,$$
where $k_{B}$ denotes Boltzmann’s constant. The basis states employed in the trace correspond to the eigenstates of the Hamiltonian $\hat{H}$, which are presented by Equation (Equation 8). Here, $\hat{H}$ signifies the Hamiltonian of the boson system and is presented by Equation (Equation 6). The thermodynamic potential is linked to the Hamiltonian of the boson system via the grand partition function,
(10)$$\Omega \left(T,\mu \right)=-k_{B}T\ln Z\;.$$
All the macroscopical thermodynamic functions may be calculated from the thermodynamic potential.

Substituting Equations (5) and (6) into Equation (Equation 9), the grand partition function is acquired as
(11)$$Z=\prod_{n_{x}=0}^{\infty }\prod_{n_{y}=0}^{\infty }\mathrm{Tr}\;\exp\left[-\left(E_{n_{x}n_{y}}-\mu \right){\hat{b}}_{n_{x}n_{y}}^{†}{\hat{b}}_{n_{x}n_{y}}/k_{B}T\right]\;.$$

If the trace in Equation (Equation 11) is written out at length with the complete set of eigenstates $|N_{n_{x}n_{y}}\rangle$ of number operator ${\hat{N}}_{n_{x}n_{y}}$, one obtains
$$\mathrm{Tr}\;\exp[-\left(E_{n_{x}n_{y}}-\mu \right){\hat{b}}_{n_{x}n_{y}}^{†}{\hat{b}}_{n_{x}n_{y}}/k_{B}T]$$
(12)$$={\left\{1-\exp[-\left(E_{n_{x}n_{y}}-\mu \right)/k_{B}T]\right\}}^{-1}\;.$$

Consequently, the grand partition function can be acquired as
(13)$$Z=\prod_{n_{x}=0}^{\infty }\prod_{n_{y}=0}^{\infty }{\left\{1-\exp[-\left(E_{n_{x}n_{y}}-\mu \right)/k_{B}T]\right\}}^{-1}\;.$$

If we take the logarithm of Equation (Equation 13), we can obtain the thermodynamic potential as
(14)$$\Omega =k_{B}T\sum_{n_{x}=0}^{\infty }\sum_{n_{y}=0}^{\infty }\ln\left\{1-\exp[-\left(E_{n_{x}n_{y}}-\mu \right)/k_{B}T]\right\}\;.$$

## 3. Thermodynamic Properties of the Ideal Boson System

### 3.1. Ushering of the Reduced Chemical Potential $x_{a}$

For temperature *T*, $\langle N_{n_{x}n_{y}}\rangle$ signifies the thermal average of the number of atoms occupying the oscillator level with $n_{x}$ and $n_{y}$. The mean occupation number $\langle N_{n_{x}n_{y}}\rangle$ of oscillator levels may be acquired from the thermodynamic potential as
(15)$$\langle N_{n_{x}n_{y}}\rangle ={\left(\frac{\partial \Omega }{\partial E_{n_{x}n_{y}}}\right)}_{T\mu }\;.$$

Substituting Equation (Equation 14) into Equation (Equation 15), one immediately finds that
(16)$$\langle N_{n_{x}n_{y}}\rangle =\frac{1}{e^{\left(E_{n_{x}n_{y}}-\mu \right)/k_{B}T}-1}\;.$$

Equation (Equation 16) represents the famous Bose–Einstein distribution. The chemical potential μ is decided by the limitation that the total number of bosons in the system equals *N*:(17)$$\sum_{n_{x}n_{y}}\langle N_{n_{x}n_{y}}\rangle =N\;.$$

In the present paper, we assume that $\omega_{x}\geq \omega_{y}$. It is necessary to define an anisotropy parameter *p* by the relation $p=\omega_{y}/\omega_{x}$. The anisotropy parameter varies in the range of 0≤p≤1. When p=0, the trap is a 1D harmonic trap along the *x* axis. When p=1, the trap is a 2D isotropic harmonic trap.

To decide μ, one must usher the fugacity *z* by the relation $z=\exp(\mu^{*}/k_{B}T)$, where we have ushered an effective chemical potential $\mu^{*}=\mu -\frac{1}{2}\hbar \left(\omega_{x}+\omega_{y}\right)$. Further, we can usher the parameters $q_{x}=\exp\left(-\hbar \omega_{x}/k_{B}T\right)$ and $x_{a}=1-\mu^{*}/\hbar \omega_{x}$. As the temperature *T* occurs in the definition $z=\exp(\mu^{*}/k_{B}T)$, the fugacity *z* can not embody the chemical potential $\mu^{*}$ by much and, hence, *z* is a bad physical quantity. The quantity $x_{a}$ embodies the chemical potential $\mu^{*}$ a lot and, therefore, $x_{a}$ corresponds to a good physical quantity. Consequently, the quantity $x_{a}$ is known as the reduced chemical potential. In the same method, the quantity $q_{x}$ embodies the temperature *T* a lot and, hence, $q_{x}$ corresponds to a good physical quantity. By means of the good physical quantities $x_{a}$ and $q_{x}$, Equation (Equation 17) is adapted into an equation of state:(18)$$\frac{q_{x}^{x_{a}-1}}{1-q_{x}^{x_{a}-1}}+H_{q_{x}}\left(x_{a}\right)=N\;,$$
where the first term signifies the number of condensed bosons and $H_{q_{x}}\left(x_{a}\right)$ represents the number of uncondensed bosons and is presented by
(19)$$H_{q_{x}}\left(x_{a}\right)=F_{q_{x}}\left(x_{a}\right)+G_{q_{x}}\left(x_{a}\right)\;,$$
(20)$$F_{q_{x}}\left(x_{a}\right)=\frac{\ln\left(1-q_{x}\right)+\psi_{q_{x}}\left(x_{a}\right)}{\ln q_{x}}\;,$$
(21)$$G_{q_{x}}\left(x_{a}\right)=\frac{k_{c}\ln\left(1-q_{x}\right)}{\ln q_{x}}+\frac{1}{\ln q_{x}}\sum_{k=1}^{k_{c}}\psi_{q_{x}}\left(x_{a}-1+kp\right)\;,$$
where the sum upper limit *∞* is substituted for an upper cutoff $k_{c}$ and, in reality, one can let $k_{c}=200$. The numeral calculation displays that the upper cutoff $k_{c}=200$ is enough for a high-precision computation. $\psi_{q}\left(x\right)$ stands for the *q*-digamma function defined by $\psi_{q}\left(x\right)=d\left[\ln\Gamma_{q}\left(x\right)\right]/dx$, where $\Gamma_{q}\left(x\right)$ stands for the *q*-gamma function defined by
(22)$$\Gamma_{q}\left(x\right)={\left(1-q\right)}^{1-x}\prod_{n=0}^{\infty }\frac{1-q^{n+1}}{1-q^{n+x}}\;,$$
when |q|<1 and x≠0, −1, −2, ⋯. Jackson ushered the *q*-gamma function [25] and Krattenthaler and Srivastava ushered the *q*-digamma function [26]. In past decades, the *q*-gamma function and the *q*-polygamma function have played an important role in science and technology [27]. The reduced chemical potential $x_{a}$ can be calculated numerically from Equation (Equation 18). $x_{a}$ is a function of temperature *T*, boson number *N*, and anisotropy parameter *p*. To meet Equation (Equation 18), we require that $x_{a}\geq 1$. When $x_{a}=1$, a 2D atomic gas goes into the state of BEC.

### 3.2. Internal Energy

In the following, we shall investigate the thermodynamic properties of ideal bosons in a 2D anisotropic harmonic potential. A major thermodynamic function in the ideal boson system corresponds to the internal energy *E*, as defined by
(23)$$E=\sum_{n_{x}n_{y}}E_{n_{x}n_{y}}\langle N_{n_{x}n_{y}}\rangle \;.$$

The internal energy *E* is decided collectively by Equations (16), (18), and (23). To calculate *E*, we first write out $E_{n_{x}n_{y}}=n_{x}\hbar \omega_{x}+n_{y}\hbar \omega_{y}+\frac{1}{2}\hbar \left(\omega_{x}+\omega_{y}\right)$. On substituting the last equation into Equation (Equation 16), Equation (Equation 16) can be adapted as follows:(24)$$\langle N_{n_{x}n_{y}}\rangle =\frac{ze^{-\beta (n_{x}\hbar \omega_{x}+n_{y}\hbar \omega_{y})}}{1-ze^{-\beta (n_{x}\hbar \omega_{x}+n_{y}\hbar \omega_{y})}}\;,$$
where $\beta =1/k_{B}T$. The system tends to go into the ground state with $n_{x}=n_{y}=0$. Because we have shifted the zero-point energy into the effective chemical potential, the ground-state energy has been taken to be zero. Substituting Equation (Equation 24) into Equation (Equation 23), we thus acquire
$$E=\sum_{n_{x}=1}^{\infty }\sum_{n_{y}=0}^{\infty }\frac{n_{x}\hbar \omega_{x}ze^{-\beta (n_{x}\hbar \omega_{x}+n_{y}\hbar \omega_{y})}}{1-ze^{-\beta (n_{x}\hbar \omega_{x}+n_{y}\hbar \omega_{y})}}$$
$$+\sum_{n_{y}=1}^{\infty }\sum_{n_{x}=0}^{\infty }\frac{n_{y}\hbar \omega_{y}ze^{-\beta (n_{x}\hbar \omega_{x}+n_{y}\hbar \omega_{y})}}{1-ze^{-\beta (n_{x}\hbar \omega_{x}+n_{y}\hbar \omega_{y})}}$$
$$=-\sum_{j=1}^{\infty }\frac{z^{j}}{j(1-q_{y}^{j})}\frac{\partial }{\partial \beta }\left(\frac{1}{1-e^{-\beta j\hbar \omega_{x}}}\right)$$
(25)$$-\sum_{j=1}^{\infty }\frac{z^{j}}{j(1-q_{x}^{j})}\frac{\partial }{\partial \beta }\left(\frac{1}{1-e^{-\beta j\hbar \omega_{y}}}\right)\;,$$
where $q_{y}=\exp\left(-\beta \hbar \omega_{y}\right)$. Notice that *z* and β are two independent Lagrange multipliers. Consequently, we keep *z* constant for the partial derivative of the brace with respect to β. Equation (Equation 25) suggests that the condensed bosons have no donation to the internal energy *E*. Finishing the partial derivative with respect to β, one can simplify Equation (Equation 25) into the following form:(26)$$E=\sum_{j=1}^{\infty }\frac{q_{x}^{jx_{a}}}{\left(1-q_{x}^{j}\right)\left(1-q_{y}^{j}\right)}\left[\frac{\hbar \omega_{x}}{\left(1-q_{x}^{j}\right)}+\frac{q_{x}^{-j}q_{y}^{j}\hbar \omega_{y}}{\left(1-q_{y}^{j}\right)}\right]\;,$$
where we have employed the relation $z=\exp(\beta \mu^{*})$ and $x_{a}=1-\mu^{*}/\hbar \omega_{x}$. The infinite series in Equation (Equation 26) converges very rapidly and, therefore, it can be computed numerically.

We can do a numeral computation about *E* and, therefore, we let $\omega_{y}/2\pi =10.0$ Hz, which is available to an actual experiment [28]. For the sake of contrast, one can usher a 2D condensation temperature $T_{2D}$ by the definition $k_{B}T_{2D}=\hbar \omega_{y}$. Consequently, we acquire $T_{2D}=0.4799$ nK. In the quantum statistical mechanics, the temperature of a 2D boson gas is meaningful only when the boson number is large enough. The internal energy *E* is a function of temperature *T*, boson number *N*, and anisotropy parameter *p*. When calculating *E*, we must combine Equation (Equation 18) with Equation (Equation 26). The reduced chemical potential $x_{a}$ can be calculated from Equation (Equation 18) to obtain the internal energy. According to Equation (Equation 26), in Figure 1 we show changing of the internal energy *E* with the boson number *N* for various *T*. We select the anisotropy parameter p=0.5. One can find that, at T=1 nK, $E/\hbar \omega_{x}\approx 7.580$ for all *N*. Put differently, when p=0.5 and T=1 nK, a 2D boson gas is always in the state of BEC for arbitrary boson number *N*. An interesting discovery is that, when T≤100 nK and $N\geq 3\times 10^{4}$, $E/\hbar \omega_{x}$ is a constant dependent of temperature. Put differently, when T≤100 nK and $N\geq 3\times 10^{4}$, a 2D boson gas is always in the state of BEC. These findings present an insight that, for a fixed *T*, there is a critical boson number $N_{c}$, above which $x_{a}=1$. Based on Equation (Equation 26), in Figure 2 we reveal changing of the internal energy *E* with the temperature *T* for various *N*. We select the anisotropy parameter p=0.5. When *N* is fixed, *E* is a monotonically increasing function of temperature *T*. When *N* is fixed, however, changing with temperature of the internal energy of a finite number of ideal bosons possesses an inflection point. The calculation shows that the inflection point corresponds to a critical temperature $T_{c}$. Figure 2 reveals that the internal energy of a finite number of ideal bosons possesses a classic limit $E=2Nk_{B}T$ as $T\gg T_{c}$. The computation also shows that, when p<0.5 and for a fixed *N*, the inflection point on the curve of *E* versus *T* disappears. According to Equation (Equation 26), in Figure 3 we display the variation of the internal energy *E* with the anisotropy parameter *p* for various *T*. In Figure 3, we take $N=10^{4}$. An interesting discovery is that, for a fixed *N* and when *T* is finite, E≠0 at p→0. When *N* and *T* are fixed, *E* is a monotonically increasing function of anisotropy parameter *p*. These findings present an insight that, when *T* and *N* are fixed, there is a critical anisotropy parameter $p_{c}$, below which $x_{a}=1$.

### 3.3. Entropy and Helmholtz Free Energy

In statistical mechanics, entropy (conventional sign *S*) is commonly interpreted as a degree of disorder. The entropy *S* of the ideal boson system can be derived from the thermodynamic potential as
(27)$$S=-{\left(\frac{\partial \Omega }{\partial T}\right)}_{\mu }\;.$$

Substituting Equation (Equation 14) into Equation (Equation 27), we immediately discover that
$$S=\frac{1}{T}\sum_{n_{x},n_{y}=0}^{\infty }\left(E_{n_{x}n_{y}}-\mu^{*}\right)\frac{1}{e^{\left(E_{n_{x}n_{y}}-\mu^{*}\right)/k_{B}T}-1}$$
(28)$$-k_{B}\sum_{n_{x},n_{y}=0}^{\infty }\ln\left\{1-\exp[-\left(E_{n_{x}n_{y}}-\mu^{*}\right)/k_{B}T]\right\}\;,$$
where we have ushered an effective chemical potential $\mu^{*}=\mu -\frac{1}{2}\hbar \left(\omega_{x}+\omega_{y}\right)$ and, therefore, $E_{n_{x}n_{y}}=n_{x}\hbar \omega_{x}+n_{y}\hbar \omega_{y}$. Utilizing Equations (16), (17), and (23), we can simplify Equation (Equation 28) as
(29)$$S/k_{B}=\frac{1}{k_{B}T}\left[E+\hbar \omega_{x}\left(x_{a}-1\right)N\right]+\sum_{j=1}^{\infty }\frac{q_{x}^{j(x_{a}-1)}}{j\left(1-q_{x}^{j}\right)\left(1-q_{y}^{j}\right)}\;,$$
where $q_{x}=\exp\left(-\beta \hbar \omega_{x}\right)$, $q_{y}=\exp\left(-\beta \hbar \omega_{y}\right)$ and we have introduced a reduced chemical potential $x_{a}=1-\mu^{*}/\hbar \omega_{x}$. *E* is the internal energy presented by Equation (Equation 26).

Now, we can evaluate the entropy *S*, which is a function of temperature *T*, boson number *N*, and anisotropy parameter *p*. If one evaluates *S*, one must associate Equation (Equation 29) with Equations (18) and (26). The reduced chemical potential $x_{a}$ can be computed from Equation (Equation 18) to acquire the internal energy and the entropy. Based on Equation (Equation 29), changing of the scaled entropy $S/k_{B}$ with the boson number *N* is presented in Figure 4 for various *T*. We select the anisotropy parameter p=0.2. Figure 4 demonstrates that, for a fixed temperature *T*, the entropy *S* is a slowly increasing function of the boson number *N* when $N>N_{c}$. According to Equation (Equation 29), changing of the scaled entropy $S/k_{B}$ with the temperature *T* is presented in Figure 5 for various *N*. We select the anisotropy parameter p=0.2. Figure 5 shows that, when T<0.4 nK, $S/k_{B}$ is a constant dependent of the boson number *N*. At T=4 nK, the curves corresponding to various *N* cross together. Moreover, Figure 5 reveals that, when T>4 nK, the entropy *S* is a rapidly ascending function of temperature *T* for a fixed boson number *N*. Based on Equation (Equation 29), in Figure 6 we show changing of the scaled entropy $S/k_{B}$ with the anisotropy parameter *p* for various *T*. In Figure 6, we take $N=10^{4}$. An interesting discovery is that, for all *N* and *T*, S≠0 at p→0. For a fixed *N* and *T*, *S* is a monotonically ascending function of anisotropy parameter *p*.

In the next step, we make a Legendre transformation of the internal energy *E* to the Helmholtz free energy *F*. The Helmholtz free energy *F* is defined by
(30)F=E−TS.

At constant temperature, the Helmholtz free energy is minimized in thermal equilibrium. Putting Equation (Equation 29) into Equation (Equation 30), one immediately finds that
(31)$$F=-\hbar \omega_{x}\left(x_{a}-1\right)N-k_{B}T\sum_{j=1}^{\infty }\frac{q_{x}^{j(x_{a}-1)}}{j\left(1-q_{x}^{j}\right)\left(1-q_{y}^{j}\right)}\;.$$

The Helmholtz free energy *F* given by Equation (Equation 31) is a function of temperature *T*, boson number *N*, and anisotropy parameter *p*. If one evaluates *F*, one must associate Equation (Equation 18) with Equation (Equation 31). The reduced chemical potential $x_{a}$ can be computed from Equation (Equation 18) to acquire the Helmholtz free energy. According to Equation (Equation 31), changing of the scaled free energy $-F/\hbar \omega_{x}$ with the boson number *N* is presented in Figure 7 for various *T*. We select the anisotropy parameter p=0.2. Figure 7 demonstrates that the free energy *F* is always negative and is a slowly descending function of the boson number *N* for a fixed temperature *T*. Based on Equation (Equation 31), changing of the scaled free energy $-F/\hbar \omega_{x}$ with the temperature *T* is presented in Figure 8 for various *N*. We select the anisotropy parameter p=0.2. Figure 8 reveals that the free energy *F* is a rapidly descending function of temperature *T* for a fixed boson number *N*. Notice that F=0 at T=0 K. At T=5 nK, the curves corresponding to various *N* cross together. According to Equation (Equation 31), in Figure 9 we show changing of the scaled free energy $-F/\hbar \omega_{x}$ with the anisotropy parameter *p* for various *T*. In Figure 9, we take $N=10^{4}$. An interesting discovery is that, for all *N* and *T*, F≠0 at p→0. When *N* and *T* are fixed, *F* is a monotonically descending function of anisotropy parameter *p*.

### 3.4. Heat Capacity at Constant Number

The heat capacity is the quantity of heat it takes to lift the temperature of a substance by one degree centigrade. This obviously depends on the conditions under which the heating happens. Since the internal energy *E* is a function of temperature *T*, boson number *N* and anisotropy parameter *p*, we study the heat capacity at constant number, which is specified by
(32)$$C_{N}\left(T\right)={\left(\frac{\partial E}{\partial T}\right)}_{N}={\left(\frac{\partial E}{\partial T}\right)}_{x_{a}}+{\left(\frac{\partial E}{\partial x_{a}}\right)}_{T}{\left(\frac{\partial x_{a}}{\partial T}\right)}_{N}\;.$$

The internal energy *E* given by Equation (Equation 26) is a composite function of intermediate variables $q_{x}$, $q_{y}$, and $x_{a}$, where $q_{x}=\exp\left(-\hbar \omega_{x}/k_{B}T\right)$ and $q_{y}=\exp\left(-\hbar \omega_{y}/k_{B}T\right)$. The reduced chemical potential $x_{a}$ is an implicit function of temperature *T*, boson number *N*, and anisotropy parameter *p* and is determined by Equation (Equation 18).

Employing Equation (Equation 26), one can readily derive the two partial derivatives of *E* as follows:(33)$$\frac{1}{k_{B}}{\left(\frac{\partial E}{\partial T}\right)}_{x_{a}}={\left(\frac{\hbar \omega_{x}}{k_{B}T}\right)}^{2}\left(S_{1}+S_{2}\right)\;,$$
$$\frac{1}{k_{B}}{\left(\frac{\partial E}{\partial x_{a}}\right)}_{T}=-T{\left(\frac{\hbar \omega_{x}}{k_{B}T}\right)}^{2}\sum_{j=1}^{\infty }\frac{jq_{x}^{jx_{a}}}{\left(1-q_{x}^{j}\right)\left(1-q_{y}^{j}\right)}$$
(34)$$\times \left[\frac{1}{\left(1-q_{x}^{j}\right)}+\frac{q_{y}^{j}\omega_{y}}{\left(1-q_{y}^{j}\right)q_{x}^{j}\omega_{x}}\right]\;,$$
where
$$S_{1}=\sum_{j=1}^{\infty }\frac{jq_{x}^{jx_{a}}}{\left(1-q_{x}^{j}\right)\left(1-q_{y}^{j}\right)}\left[\frac{1}{\left(1-q_{x}^{j}\right)}+\frac{q_{y}^{j}\omega_{y}}{\left(1-q_{y}^{j}\right)q_{x}^{j}\omega_{x}}\right]$$
(35)$$\times \left[x_{a}+\frac{q_{x}^{j}}{\left(1-q_{x}^{j}\right)}+\frac{q_{y}^{j}}{\left(1-q_{y}^{j}\right)}\right]\;,$$
$$S_{2}=\sum_{j=1}^{\infty }\frac{jq_{x}^{jx_{a}}}{\left(1-q_{x}^{j}\right)\left(1-q_{y}^{j}\right)}$$
(36)$$\times \left[\frac{q_{x}^{j}}{{\left(1-q_{x}^{j}\right)}^{2}}-\frac{q_{y}^{j}\omega_{y}}{\left(1-q_{y}^{j}\right)q_{x}^{j}\omega_{x}}+\frac{q_{y}^{j}\omega_{y}^{2}}{{\left(1-q_{y}^{j}\right)}^{2}q_{x}^{j}\omega_{x}^{2}}\right]\;.$$

If we utilize Equation (Equation 18) and make some arrangements, then we can infer the partial derivative of $x_{a}$ as follows:(37)$${\left(\frac{\partial x_{a}}{\partial T}\right)}_{N}=\frac{S_{4}+S_{5}+S_{6}+S_{7}+\frac{\left(x_{a}-1\right)q_{x}^{x_{a}-1}}{{\left(1-q_{x}^{x_{a}-1}\right)}^{2}}+x_{a}{\left(\frac{k_{B}T}{\hbar \omega_{x}}\right)}^{2}\psi_{q_{x}}^{\prime }\left(x_{a}\right)}{T\left[\frac{q_{x}^{x_{a}-1}}{{\left(1-q_{x}^{x_{a}-1}\right)}^{2}}+{\left(\frac{k_{B}T}{\hbar \omega_{x}}\right)}^{2}\psi_{q_{x}}^{\prime }\left(x_{a}\right)+S_{3}\right]}\;,$$
where $\psi_{q}^{\prime }\left(x\right)=d\left[\psi_{q}\left(x\right)\right]/dx$ denotes the *q*-trigamma function and
(38)$$S_{3}=\sum_{j=1}^{\infty }\frac{jq_{x}^{j(x_{a}-1)}q_{y}^{j}}{\left(1-q_{x}^{j}\right)\left(1-q_{y}^{j}\right)}\;,$$
(39)$$S_{4}=\sum_{j=1}^{\infty }\frac{jq_{x}^{j(x_{a}+1)}}{{\left(1-q_{x}^{j}\right)}^{2}}\;,$$
(40)$$S_{5}=\left[\left(x_{a}-1\right)+\left(\frac{\omega_{y}}{\omega_{x}}\right)\right]S_{3}\;,$$
(41)$$S_{6}=\sum_{j=1}^{\infty }\frac{jq_{x}^{jx_{a}}q_{y}^{j}}{{\left(1-q_{x}^{j}\right)}^{2}\left(1-q_{y}^{j}\right)}\;,$$
(42)$$S_{7}=\left(\frac{\omega_{y}}{\omega_{x}}\right)\sum_{j=1}^{\infty }\frac{jq_{x}^{j(x_{a}-1)}q_{y}^{2j}}{\left(1-q_{x}^{j}\right){\left(1-q_{y}^{j}\right)}^{2}}\;.$$

The evaluation displays that the system possesses a critical boson number $N_{c}$ at which the heat capacity at constant number reaches the maximum value. An interesting discovery is that, in Equation (Equation 32), ${\left(\partial E/\partial T\right)}_{x_{a}}$ is positive but ${\left(\partial E/\partial x_{a}\right)}_{T}$ is negative. ${\left(\partial E/\partial T\right)}_{x_{a}}$ signifies the effect of temperature, which damages BEC, while ${\left(\partial E/\partial x_{a}\right)}_{T}$ stands for the effect of chemical potential, which keeps BEC. Consequently, there is a contest mechanism between the first and second terms in Equation (Equation 32). If the two effects equalize, the system possesses a critical boson number $N_{c}$, below which the 2D system goes into the normal state, but above which the 2D system enters the BEC state.

Afterwards, the reduced chemical potential is employed in Equation (Equation 32) to acquire the heat capacity at constant number. $C_{N}$ is a function of temperature *T*, boson number *N*, and anisotropy parameter *p*. If one evaluates $C_{N}$, one must associate Equation (Equation 18) with Equation (Equation 32). Based on Equation (Equation 32), changing of the scaled heat capacity $C_{N}/k_{B}$ with the boson number *N* is presented in Figure 10 for various *T*. We select the anisotropy parameter p=0.1. An interesting discovery is that, at T=1 nK, $C_{N}/k_{B}\approx 26.0$ for all *N*. Figure 10 demonstrates that, for a finite number of 2D bosons, the system possesses an approximate critical boson number $N_{c}$. If *T* is fixed, the heat capacity $C_{N}$ is an ascending function of the boson number *N* when *N* is smaller than $N_{c}$. As $N\geq N_{c}$, $C_{N}/k_{B}$ corresponds to a constant dependent of *T*. According to Equation (Equation 32), changing of the scaled heat capacity $C_{N}/k_{B}$ with the temperature *T* is presented in Figure 11 for various *N*. We select the anisotropy parameter p=0.1. Obviously, $C_{N}=0$ at T=0 K. Figure 11 demonstrates that, for p=0.1 and at a fixed *N*, there is an inflection point on the curve of $C_{N}$ versus *T*, which corresponds to the critical temperature $T_{c}$. If *N* is fixed, the heat capacity $C_{N}$ is a fast ascending function of temperature *T* when *T* is smaller than $T_{c}$, but at $T\geq T_{c}$, it is a slowly increasing function of temperature *T*. In Figure 12, we also display changing of the scaled heat capacity $C_{N}/k_{B}$ with the temperature *T* for various *p* at N=1000. Figure 12 reveals that, for p≥0.5, the system possesses a critical temperature $T_{c}$ at which $C_{N}\left(T\right)$ attains a maximal value. The specific-heat maximum is a monotonically increasing function of anisotropy parameter *p*. The circumstance of phase transitions grows clearer and clearer as the anisotropy parameter approaches unity. The specific-heat maximum represents an important marker in this transition. When $T\gg T_{c}$, $C_{N}/k_{B}=2N$, which corresponds to the classic limit. According to Equation (Equation 32), in Figure 13 we show the variation of the scaled heat capacity $C_{N}/k_{B}$ with the anisotropy parameter *p* for various *N*. The temperature is fixed at T=50 nK. It is interesting to note that, for *N* = 10,000, $C_{N}$ possesses the minimum at p=0.04 and the maximum at p=0.51. For N≤1000, $C_{N}$ is a monotonically descending function of anisotropy parameter *p*. A fine character of the accurate solutions in the above is that they are effective for arbitrary *T*, *N*, and *p*.

## 4. Phase Transitions of Ideal Bosons in the Thermodynamic Limit

### 4.1. Critical Temperature and Reduced Chemical Potential

In the following, we intend to study the thermodynamic limit in the 2D anisotropic harmonic potential. At first, one must observe that the thermodynamic limit in the anisotropic potential corresponds to taking N→∞ and $\sqrt{\omega_{x}\omega_{y}}\to 0$ with the product $N\omega_{x}\omega_{y}$ kept constant [24]. Then, we can adapt Equation (Equation 18) in the form,
(43)$$N_{0}+H_{q_{x}}\left(x_{a}\right)=N\;,$$
where $N_{0}$ signifies the boson number in the ground state $n_{x}=n_{y}=0$. We all know that, as N→∞, $x_{a}=1$. It can be verified that the above program of the thermodynamic limit in the anisotropic potential amounts to letting $N_{0}=0$ and $x_{a}=1$ in Equation (Equation 43). The operation leads to the following expression for the critical temperature $T_{c}$,
(44)$$H_{q_{xc}}\left(1\right)=N\;,$$
where $q_{xc}=\exp\left(-\hbar \omega_{x}/k_{B}T_{c}\right)$. In the trap thermodynamic limit, one can derive the solution of Equation (Equation 43) as
(45)$$x_{a}=\left\{\begin{array}{c} 1\;,\;T\leq T_{c}\;,\\ \mathrm{the}\;\mathrm{root}\;\mathrm{of}\;H_{q_{x}}\left(x_{a}\right)=N\;,\;T>T_{c}\;. \end{array}\right.$$

Even though Equations (44) and (45) are acquired in the condition of N→∞, they are effective for arbitrary *T* and large *N* ($N\geq 10^{3}$). As $10^{2}\leq N<10^{3}$, Equations (44) and (45) are acceptable too. In reality, one first uses Equation (Equation 44) to resolve the transition temperature $T_{c}$ and then employs Equation (Equation 45) to resolve the reduced chemical potential $x_{a}$. The critical temperature $T_{c}$ given by Equation (Equation 44) is an implicit function of boson number *N* and anisotropy parameter *p*. As *p* is fixed, the critical temperature $T_{c}$ is a monotonically ascending function of boson number *N*. As *N* is fixed, the critical temperature $T_{c}$ is a monotonically descending function of anisotropy parameter *p*. When the number of harmonically trapped 2D bosons is finite, the properties of phase transitions at some critical temperature are very clear too. The above investigation obviously demonstrates that the properties of phase transitions in the anisotropic potential are more vivid than those in the isotropic potential.

### 4.2. Internal Energy, Entropy, and Helmholtz Free Energy

The statistical mechanics of phase transitions can portray phase transitions on the basis of thermodynamic functions. At critical temperatures, the Helmholtz free energy of the two phases should be successive. Nevertheless, phase transitions could be sorted into two kinds based on the properties of derivatives of the Helmholtz free energy. When the first derivatives of the Helmholtz free energy are not successive, the accompanied phase transitions are known as first-order phase transitions. When the second derivatives of the Helmholtz free energy are not successive, the accompanied phase transitions are known as second-order phase transitions.

In the anisotropic trap, the Helmholtz free energy *F* is a function of temperature *T*, boson number *N*, and anisotropy parameter *p*, that is, F=F(T,N,p). The entropy *S* is linked to the first derivative of *F*:(46)$$S=-{\left(\frac{\partial F}{\partial T}\right)}_{N}\;.$$

Accordingly, when first-order phase transitions occur, the entropy of two phases *a* and *b* is discontinuous, $S_{a}\neq S_{b}$. When a transition between two phases *a* and *b* happens, the latent heat *L* is presented by $L=T(S_{a}-S_{b})$. A first-order phase transition possesses two significant features: at the transition, the system has a latent heat, and the system makes a jump in the entropy. From gas to liquid phase, from liquid to solid phase, and from gas to solid phase, these transitions are all first-order phase transitions. The heat capacity $C_{N}$ at constant *N* is linked to the second derivative of *F*:(47)$$C_{N}=-T{\left(\frac{\partial^{2}F}{\partial T^{2}}\right)}_{N}\;.$$

Accordingly, when second-order phase transitions occur, the heat capacity of two phases *a* and *b* is not successive, $C_{a}\neq C_{b}$. A second-order phase transition possesses two significant features: at the transition, the system has no latent heat, and the system makes a jump in the heat capacity. When there is no magnetic field, the transition between superconductive and normal phases of a metal corresponds to a second-order phase transition.

At this point, we start to depict the transition behavior of ideal bosons in a 2D anisotropic harmonic potential. Based on Equations (26), (44), and (45), Figure 14 reveals the internal energy *E* versus the temperature *T* for $N=10^{4}$ and p=0.5. Owing to Equation (Equation 44), we discover that, at $N=10^{4}$ and p=0.5, $T_{c}=51.90$ nK. In Figure 14, as the temperature *T* traverses $T_{c}=51.90$ nK, the internal energy *E* is successive and flat. Computations also demonstrate that, at $T_{c}=51.90$ nK, both the entropy *S* and the free energy *F* are successive and flat and do not reveal any clue of phase transitions.

### 4.3. Heat Capacity at Constant Number

In what follows, we can inspect the transition properties of the heat capacity at constant number. As $T\leq T_{c}$ and since $x_{a}=1$, ${\left(\partial x_{a}/\partial T\right)}_{N}=0$. As $T>T_{c}$, ${\left(\partial x_{a}/\partial T\right)}_{N}$ is resolved by the partial derivative of the expression $H_{q_{x}}\left(x_{a}\right)=N$. As a result, we obtain the heat capacity at constant *N* as follows:(48)$$C_{N}=\left\{\begin{array}{c} {\left(\frac{\partial E}{\partial T}\right)}_{x_{a}}\;,\;T\leq T_{c}\;,\\ {\left(\frac{\partial E}{\partial T}\right)}_{x_{a}}+{\left(\frac{\partial E}{\partial x_{a}}\right)}_{T}{\left(\frac{\partial x_{a}}{\partial T}\right)}_{N}\;,\;T>T_{c}\;. \end{array}\right.$$

As $T>T_{c}$, one can acquire the partial derivative of $x_{a}$ as follows:(49)$${\left(\frac{\partial x_{a}}{\partial T}\right)}_{N}=\frac{S_{4}+S_{5}+S_{6}+S_{7}+x_{a}{\left(\frac{k_{B}T}{\hbar \omega_{x}}\right)}^{2}\psi_{q_{x}}^{\prime }\left(x_{a}\right)}{T\left[{\left(\frac{k_{B}T}{\hbar \omega_{x}}\right)}^{2}\psi_{q_{x}}^{\prime }\left(x_{a}\right)+S_{3}\right]}\;.$$

When a phase transition happens, the heat capacity of two phases *a* and *b* is not successive, $C_{a}\neq C_{b}$. Therefore, at transition temperature $T_{c}$, the system makes a jump in the heat capacity and this jump is presented by
(50)$$\Delta C_{N}=-{\left(\frac{\partial E}{\partial x_{a}}\right)}_{T}{\left(\frac{\partial x_{a}}{\partial T}\right)}_{N}\;,\;T=T_{c}\;.$$

Since, at $T_{c}$, the heat capacity of the BEC state is larger than that of the normal state, this leap takes place. As a result, the transition between normal and BEC states corresponds to a second-order phase transition.

When calculating $C_{N}$ in the thermodynamic limit, we must combine Equation (Equation 48) with Equations (44) and (45). According to Equations (44), (45), and (48), Figure 15 demonstrates the heat capacity $C_{N}$ against the temperature *T* for $N=10^{4}$ and $10^{5}$ at p=0.5. From Equation (Equation 44), we find that, at $N=10^{4}$, $10^{5}$ and p=0.5, $T_{c}=51.90$ and 166.07 nK, respectively. The cusp singularity of a second-order phase transition of BEC in 2D anisotropic traps is displayed in Figure 15. Such a critical property resembles the λ-transition of liquid helium-4. An up-to-date BEC experiment with 2D ideal photons has showed this cusp singularity of the specific heat [29]. Consequently, our program for the thermodynamic limit is totally proper. Owing to Figure 15, we can find that the heat capacity of the BEC state is extremely larger than that of the normal state. At critical temperature $T_{c}$, this leap in the heat capacity is immense. Further, Figure 16 reveals the heat capacity $C_{N}$ against the temperature *T* for $N=10^{4}$ and p=0.3. Owing to Equation (Equation 44), one can discover that, at $N=10^{4}$ and p=0.3, $T_{c}=66.69$ nK. Comparing Figure 15 with Figure 16, one can see that, for a fixed *N* and when *p* is lowered, the singular peak of the specific heat becomes very gentle. Many properties of BEC of 2D anisotropic traps are exhibited in the present paper.

## 5. Conclusions

This thesis investigates the thermodynamic behavior of ideal bosons in 2D anisotropic harmonic traps. In this study, we find that the issue of BEC in 2D anisotropic harmonic potentials can be figured out analytically. The analytical solution is connected with an analytical expression, which includes a series expansion of *q*-digamma functions. The *q*-digamma function was ushered in mathematics thirty years before and, at present, plays an important role in science and technology. For this probe, we usher a new thermodynamic function that is known as the reduced chemical potential to substitute the fugacity. In the issue of BEC of harmonically trapped 2D bosons, the fugacity *z* is a bad thermodynamic function, but the reduced chemical potential $x_{a}$ is a fine thermodynamic function. We construct a hypothetical model of quantum thermodynamics to evaluate various thermodynamic functions of ideal bosons in 2D anisotropic harmonic traps. These thermodynamic quantities are functions of temperature *T*, boson number *N*, and anisotropy parameter *p*. The graphs of their variation with *T*, *N*, and *p* are given.

An interesting observation is that, when *T* and *N* are fixed, the entropy of a finite number of ideal bosons is a monotonically ascending function of anisotropy parameter *p* but the free energy is a monotonically descending function of anisotropy parameter *p*. The first major discovery is that the internal energy of a finite number of ideal bosons is a monotonically ascending function of anisotropy parameter *p*. The second major discovery is that, when p≥0.5, the changing with temperature of the heat capacity of a finite number of ideal bosons possesses the maximum value, which happens at critical temperature $T_{c}$. The third major discovery is that, when 0.1≤p<0.5, the changing with temperature of the heat capacity of a finite number of ideal bosons possesses an inflection point, but when p<0.1, the inflection point disappears. The fourth main finding is that, for a fixed *T* and when $N<10^{4}$, the heat capacity of a finite number of ideal bosons is a monotonically descending function of anisotropy parameter *p*, but when $N\geq 10^{4}$, $C_{N}$ possesses a minimum and a maximum. Therefore, a finite number of ideal bosons in 2D anisotropic harmonic traps possess larger heat capacity $C_{N}$ than those in 2D isotropic harmonic traps. The fifth major discovery is that, in the thermodynamic limit, at $T_{c}$ and when p≥0.5, the heat capacity at constant number reveals a cusp singularity, which resembles the λ-transition of liquid helium-4. The sixth major discovery is that, in comparison to 2D isotropic harmonic traps (p=1), the singular peak of the specific heat becomes very gentle when *p* is lowered. Since the observation of the BEC of ultracold dilute atomic gases in 1995, one has thought that the thermodynamic properties of 2D anisotropic harmonic traps can be evaluated exactly. This paper accomplishes this task successfully.

This thesis has studied phase transitions of ideal bosons in 2D anisotropic harmonic traps in the thermodynamic limit. Nevertheless, one can find that, when the boson number in 2D anisotropic harmonic traps is finite, the properties of phase transitions at some critical temperature are very clear as well. This thesis demonstrates that, in the trap thermodynamic limit, the system possesses an accurate critical temperature $T_{c}$ settled by a strict expression, which includes a series expansion of *q*-digamma functions. The calculation displays that, when p≥0.5, the 2D critical temperature $T_{c}$ is identical to the specific-heat maximum of a finite number of harmonically trapped 2D bosons. Calculations also display that, at $T_{c}$ both the entropy *S* and the free energy *F* are successive and flat and do not reveal any clue of phase transitions. However, one can discover that, at $T_{c}$, the heat capacity at constant number is not successive. Therefore, at $T_{c}$, the system possesses a leap in the heat capacity at constant number. The leap in the heat capacity at constant number is enormous. Since, at $T_{c}$, the heat capacity of the BEC state is much larger than that of the normal state, this leap takes place. The dissection of 2D anisotropic harmonic traps sheds more light on the behavior of BEC.

To sum up, the boson system in the BEC state is comprised of condensed bosons and uncondensed bosons. We ascertain that, in 2D anisotropic traps, the number of uncondensed bosons is represented by an analytical expression, which includes a series expansion of *q*-digamma functions in mathematics. One can utilize this analytic result to evaluate various thermodynamic functions of ideal bosons in 2D anisotropic harmonic traps. The first major discovery is that, in the thermodynamic limit, at $T_{c}$ and when p≥0.5, the heat capacity at constant number reveals a cusp singularity, which resembles the λ-transition of liquid helium-4. Another major discovery is that, in comparison to 2D isotropic harmonic traps (p=1), the singular peak of the specific heat becomes very gentle when *p* is lowered. In this probe, we usher a new thermodynamic function that is known as the reduced chemical potential to substitute the fugacity.

## Figures and Tables

**Figure 1 entropy-25-01513-f001:**
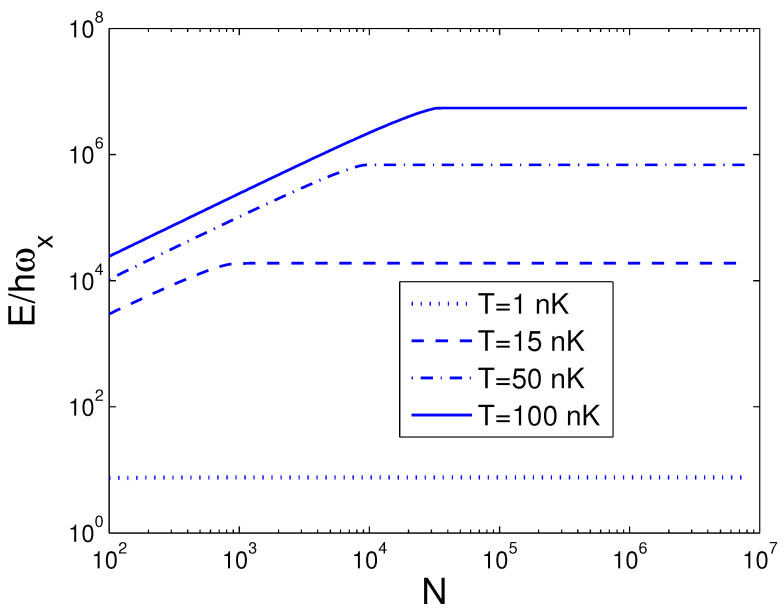
A plot of the scaled internal energy $E/\hbar \omega_{x}$ versus the boson number *N* in a 2D ideal boson gas. The plot is drawn for T=1, 15, 50, 100 nK. We select the axial angular frequency $\omega_{y}/2\pi =10.0$ Hz and the anisotropy parameter p=0.5.

**Figure 2 entropy-25-01513-f002:**
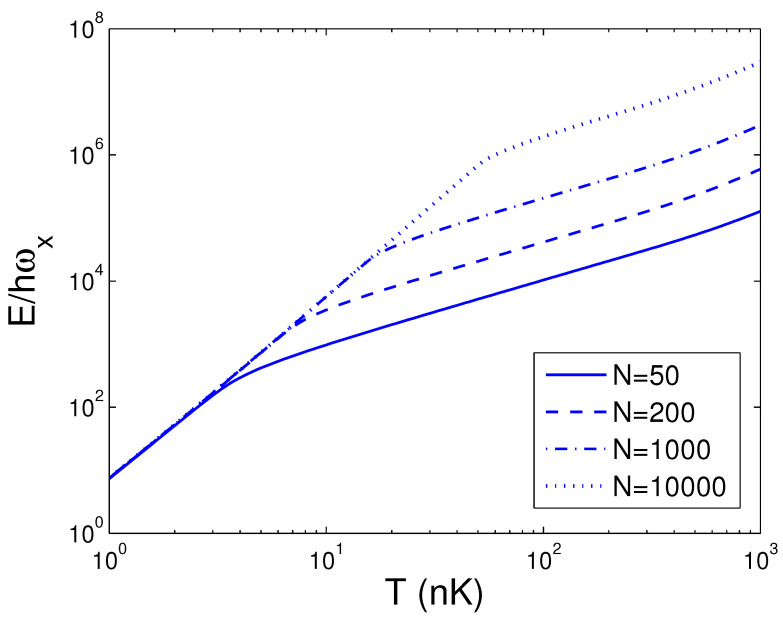
A plot of the scaled internal energy $E/\hbar \omega_{x}$ versus the temperature *T* in a 2D ideal boson gas. The plot is drawn for N=50, 200, 1000, 10,000. We select the axial angular frequency $\omega_{y}/2\pi =10.0$ Hz and the anisotropy parameter p=0.5.

**Figure 3 entropy-25-01513-f003:**
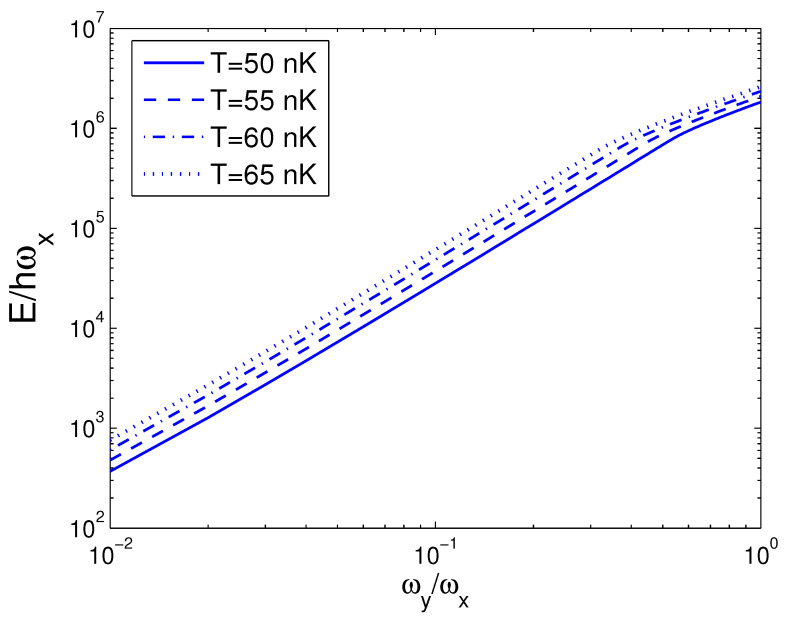
A plot of the scaled internal energy $E/\hbar \omega_{x}$ versus the anisotropy parameter *p* in a 2D ideal boson gas. The plot is drawn for T=50, 55, 60, 65 nK and $N=10^{4}$. We select the axial angular frequency $\omega_{y}/2\pi =10.0$ Hz.

**Figure 4 entropy-25-01513-f004:**
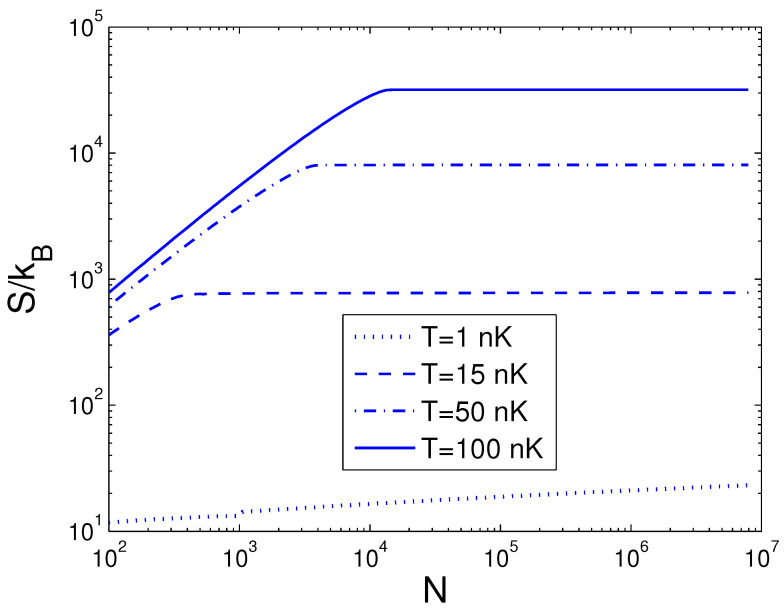
A plot of the scaled entropy $S/k_{B}$ versus the boson number *N* in a 2D ideal boson gas. The plot is drawn for T=1, 15, 50, 100 nK. We select the axial angular frequency $\omega_{y}/2\pi =10.0$ Hz and the anisotropy parameter p=0.2.

**Figure 5 entropy-25-01513-f005:**
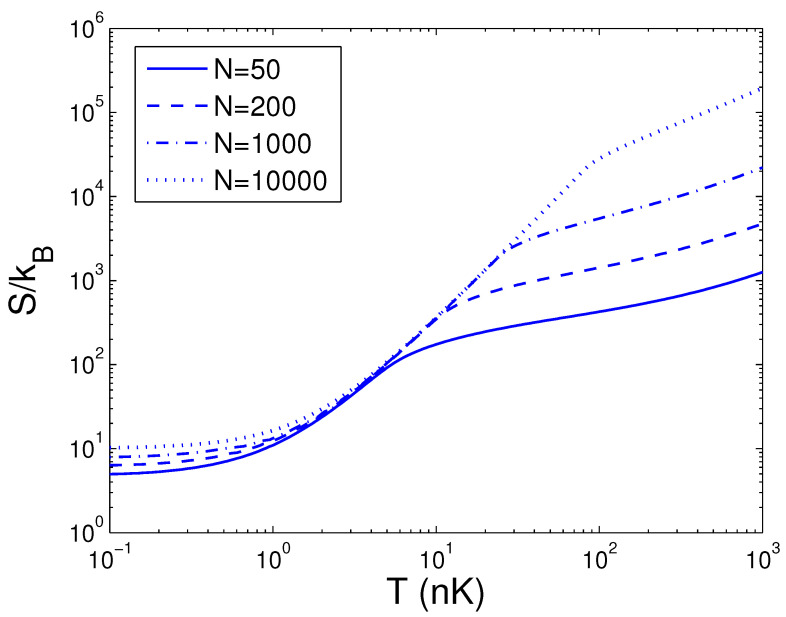
A plot of the scaled entropy $S/k_{B}$ versus the temperature *T* in a 2D ideal boson gas. The plot is drawn for N=50, 200, 1000, 10,000. We select the axial angular frequency $\omega_{y}/2\pi =10.0$ Hz and the anisotropy parameter p=0.2.

**Figure 6 entropy-25-01513-f006:**
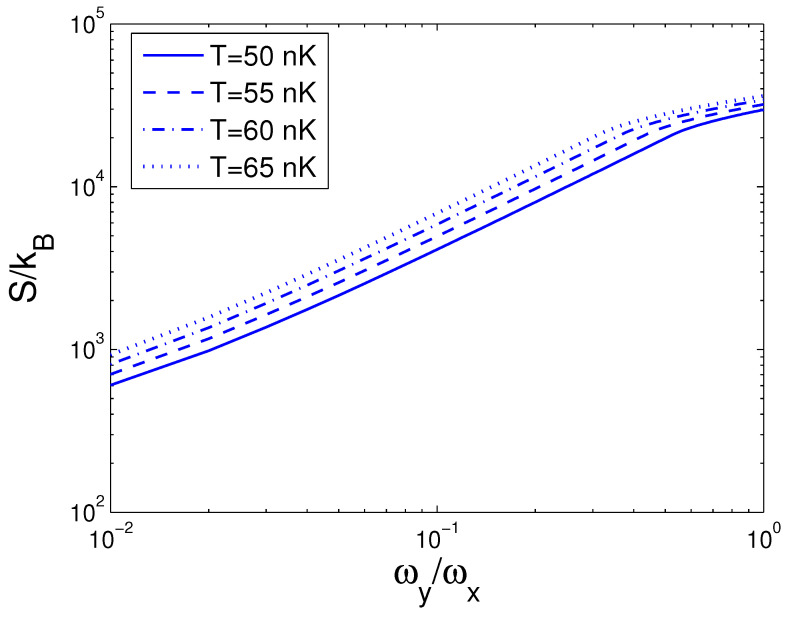
A plot of the scaled entropy $S/k_{B}$ versus the anisotropy parameter *p* in a 2D ideal boson gas. The plot is drawn for T=50, 55, 60, 65 nK and $N=10^{4}$. We select the axial angular frequency $\omega_{y}/2\pi =10.0$ Hz.

**Figure 7 entropy-25-01513-f007:**
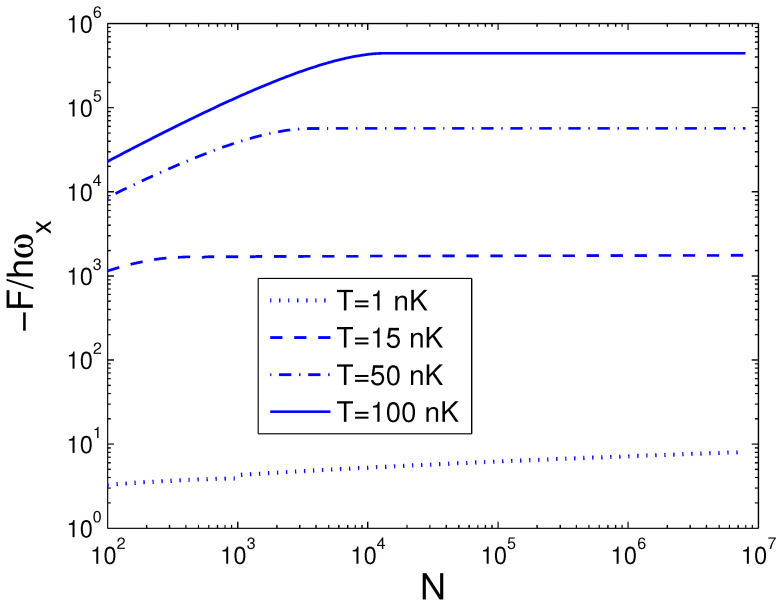
A plot of the scaled free energy $-F/\hbar \omega_{x}$ versus the boson number *N* in a 2D ideal boson gas. The plot is drawn for T=1, 15, 50, 100 nK. We select the axial angular frequency $\omega_{y}/2\pi =10.0$ Hz and the anisotropy parameter p=0.2.

**Figure 8 entropy-25-01513-f008:**
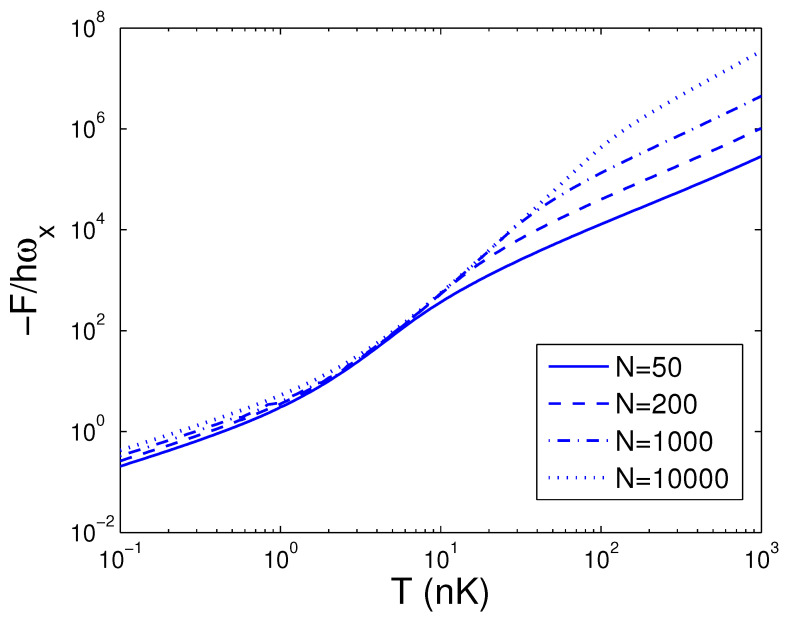
A plot of the scaled free energy $-F/\hbar \omega_{x}$ versus the temperature *T* in a 2D ideal boson gas. The plot is drawn for N=50, 200, 1000, 10,000. We select the axial angular frequency $\omega_{y}/2\pi =10.0$ Hz and the anisotropy parameter p=0.2.

**Figure 9 entropy-25-01513-f009:**
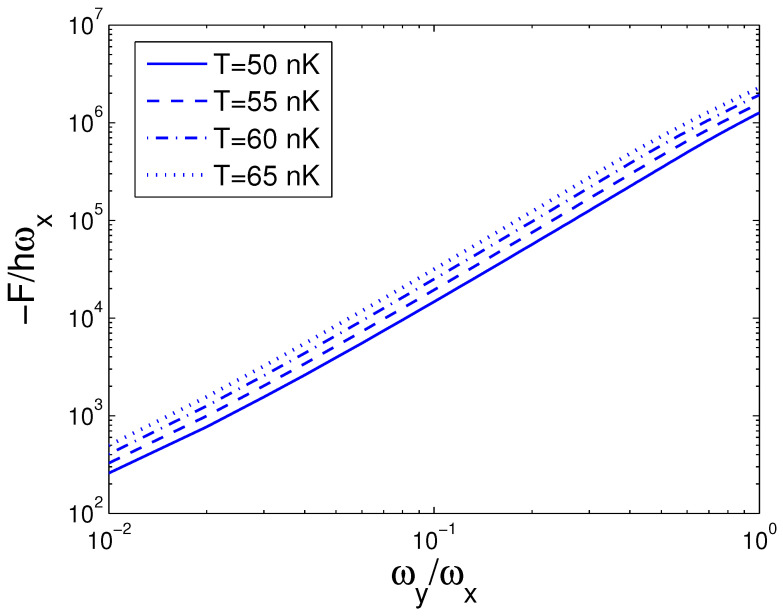
A plot of the scaled free energy $-F/\hbar \omega_{x}$ versus the anisotropy parameter *p* in a 2D ideal boson gas. The plot is drawn for T=50, 55, 60, 65 nK and $N=10^{4}$. We select the axial angular frequency $\omega_{y}/2\pi =10.0$ Hz.

**Figure 10 entropy-25-01513-f010:**
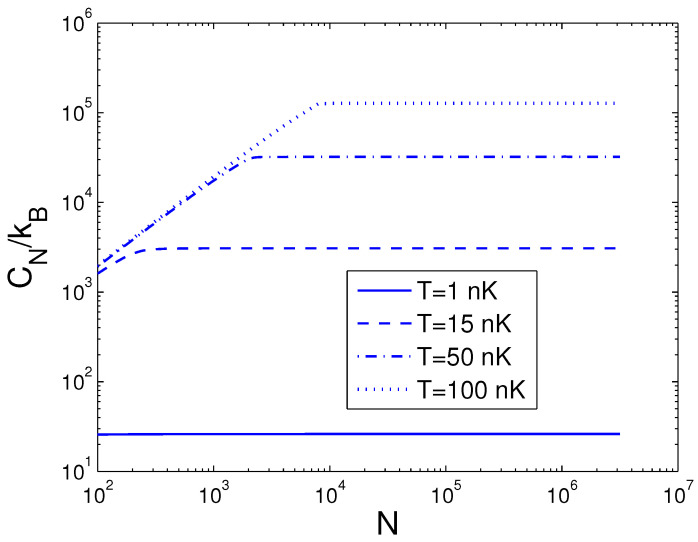
A plot of the scaled heat capacity $C_{N}/k_{B}$ versus the boson number *N* in a 2D ideal boson gas. The plot is drawn for T=1, 15, 50, 100 nK. We select the axial angular frequency $\omega_{y}/2\pi =10.0$ Hz and the anisotropy parameter p=0.1.

**Figure 11 entropy-25-01513-f011:**
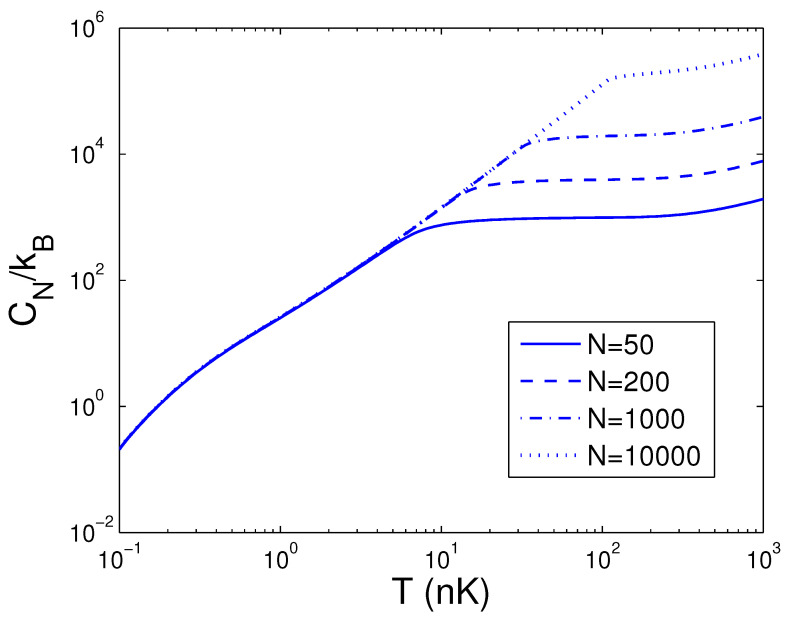
A plot of the scaled heat capacity $C_{N}/k_{B}$ versus the temperature *T* in a 2D ideal boson gas. The plot is drawn for N=50, 200, 1000, 10,000. We select the axial angular frequency $\omega_{y}/2\pi =10.0$ Hz and the anisotropy parameter p=0.1.

**Figure 12 entropy-25-01513-f012:**
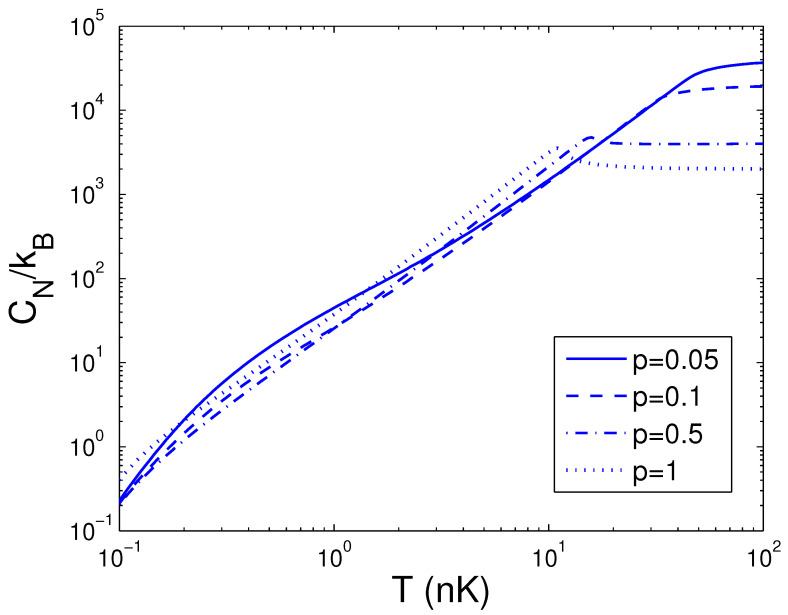
A plot of the scaled heat capacity $C_{N}/k_{B}$ versus the temperature *T* in a 2D ideal boson gas. The plot is drawn for p=0.05, 0.1, 0.5, 1. We select the axial angular frequency $\omega_{y}/2\pi =10.0$ Hz and the boson number N=1000.

**Figure 13 entropy-25-01513-f013:**
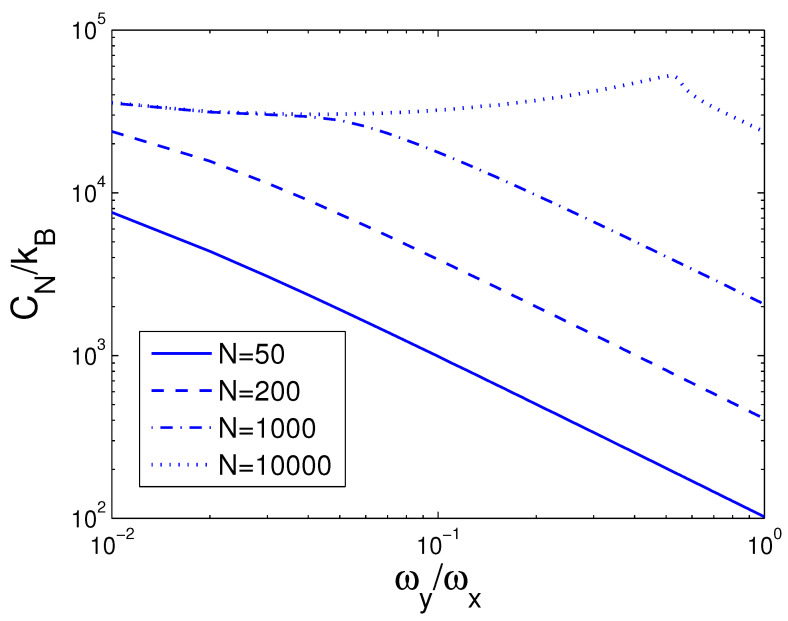
A plot of the scaled heat capacity $C_{N}/k_{B}$ versus the anisotropy parameter *p* in a 2D ideal boson gas. The plot is drawn for N=50, 200, 1000, 10,000. We select the axial angular frequency $\omega_{y}/2\pi =10.0$ Hz and the temperature T=50 nK.

**Figure 14 entropy-25-01513-f014:**
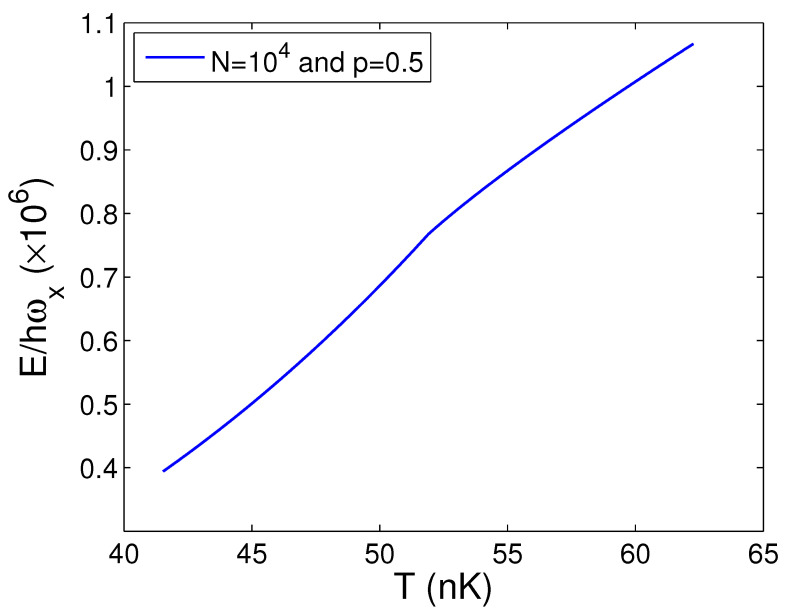
In the thermodynamic limit, changing of the scaled internal energy $E/\hbar \omega_{x}$ with the temperature *T*. We select the axial angular frequency $\omega_{y}/2\pi =10.0$ Hz, the boson number *N* = 10,000, and the anisotropy parameter p=0.5.

**Figure 15 entropy-25-01513-f015:**
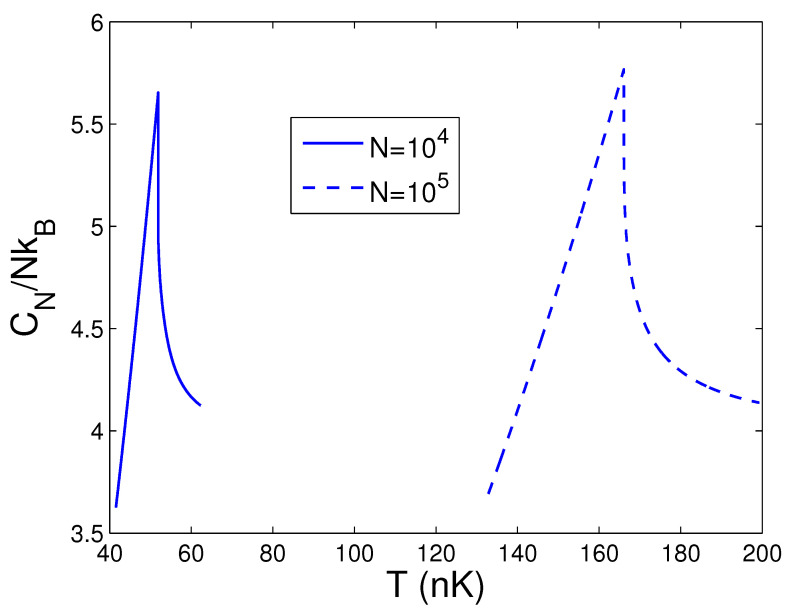
In the thermodynamic limit, changing of the scaled heat capacity $C_{N}/Nk_{B}$ with the temperature *T* at $N=10^{4}$ and $10^{5}$. We select the axial angular frequency $\omega_{y}/2\pi =10.0$ Hz and the anisotropy parameter p=0.5.

**Figure 16 entropy-25-01513-f016:**
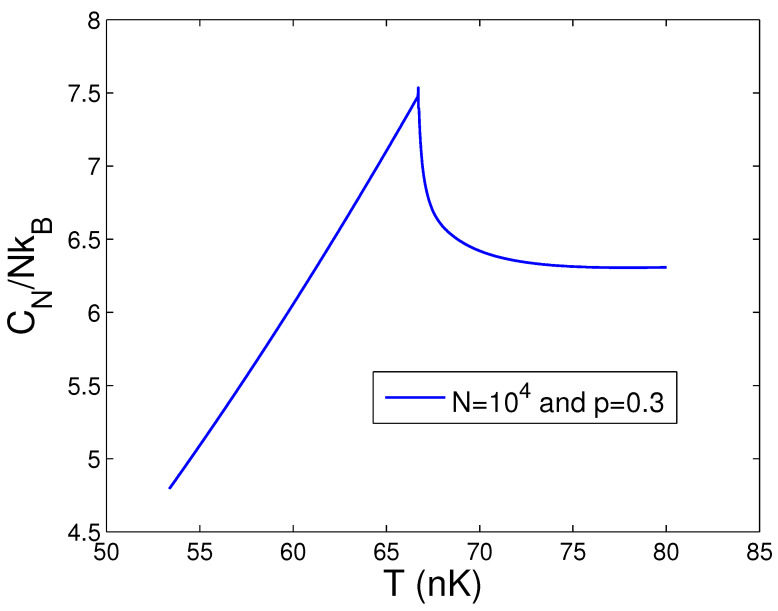
In the thermodynamic limit, changing of the scaled heat capacity $C_{N}/Nk_{B}$ with the temperature *T* at $N=10^{4}$ and p=0.3. We select the axial angular frequency $\omega_{y}/2\pi =10.0$ Hz.

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
