# Peer review of "Accurate Thermodynamic Properties of Ideal Bosons in a Highly Anisotropic 2D Harmonic Potential"

_entropy, 2023, doi:10.3390/e25111513_

Round 1

Reviewer 1 Report

The paper investigates theoretically the problem of non-interacting bosons in an anisotropic two-dimensional (2D) harmonic trap.

The main problem is that, in the Abstract, in the Introduction and in the Conclusions, it is used the symbol "p" without a clear definition. For instance, in the Abstract "p" is introduced in the 6th row and only in the 8th row is is written that "p" is the "anisotropic parameter".

By the way, "anisotropy parameter" are more appropriate words with respect the "anisotropic parameter".

In addition, in the rest of the paper it is always used "\omega_y/\omega_x" instead of "p".

Some relevant references are missing. Among them,

- Ze Cheng J. Stat. Mech. 113103 (2017)

where it is discussed the isotropic version of the present paper

- Ze Cheng and Jiang Hong Man, Canadian Journal of Physics 98, N. 2 (2020)

where it is discussed analytically the same anisotropic case of the present paper.

The author must stress the differences with respect these two previously published papers.

In addition, it is relevant to cite

- L. Salasnich, J. Math. Phys. 41, 8016–8024 (2000) 

that analyzes the thermodynamics of ideal gases in a generic power low potential

- J. Klaers, J. Schmitt, F. Vewinger, and M. Weitz, Nature 468, 545 (2010).

F. E. Ozturk, F. Vewinger, M. Weitz, and J. Schmitt, Phys. Rev. Lett. 130, 033602 (2023)

that investigate both experimentally and theoretically the statistical mechanics of a gas of "massive" photons in a 2D harmonic potential.

Reviewer 2 Report

see attached file

Round 2

Reviewer 1 Report

The paper has been modified following my suggestions. I think that now the paper could be published in Entropy.

Reviewer 2 Report

The manuscript improved during the resubmission process. In particular, the self-plagiatism is now removed. Therefore, I can now recommend publication.

The English is not perfect but acceptable for a publication.